# Synergy of metal nanoparticles and organometallic complex in NAD(P)H regeneration via relay hydrogenation

Maodi Wang[1,2], Zhenchao Zhao[3], Chunzhi Li[1,2], He Li ⓘ [1], Jiali Liu[1,2] & Qihua Yang ⓘ [3] ✉

Most, if not all, of the hydrogenation reactions are catalyzed by organometallic complexes (M) or heterogeneous metal catalysts, but to improve both the activity and selectivity simultaneously in one reaction via a rational combination of the two types of catalysts remains largely unexplored. In this work, we report a hydrogenation mode though H species relay from supported metal nanoparticles (NPs) to M, where the former is responsible for $H_2$ dissociation, and M is for further hydride transferring to reactants. The synergy between metal NPs and M yields an efficient NAD(P)H regeneration system with >99% selectivity and a magnitude higher activity than the corresponding metal NPs and M. The modularizing of hydrogenation reaction into hydrogen activation with metal NPs and substrate activation with metal complex paves a new way to rationally address the challenging hydrogenation reactions.

Selective hydrogenation is one of the most important chemical transformations in the field from petrochemical to fine chemical industries[1–4]. The ideal scenario for a given reaction would be the 100% yield of target product with low energy consumption[5,6]. In the course of more than one century of extensive investigations, two categories of catalytic processes, namely homogenous and heterogeneous catalysis[1,7,8], are established for such a goal. Correspondingly, homogenous organometallic complexes (M) and supported metal nanoparticles (NPs) are the most widely used catalysts for each process, respectively[1]. Both types of catalysts are well-developed and hold unique merits, such as the high regio/stereo-selectivity for M[9–11] and the easy recovery and strong $H_2$ dissociation ability for supported metal NPs[1,12], which make each of them irreplaceable and yet not universal[13,14]. Traditionally, homogenous and heterogeneous catalysts were applied separately and rarely catalyzed a target reaction synergistically. Very recently, Andersson's group reported that homogenous Rh complex and in situ formed Rh NPs could efficiently catalyze the cascade hydrogenation of olefin and arene in one system without interference from each other[15]. Better yet, if we could go a step further to make these two types of catalysts communicate with each other purposely via a controllable manner, it may help addressing the challenges facing homogeneous and heterogeneous catalysis, but this remains largely unexplored so far.

In situ regeneration of the reduced form cofactors nicotinamide adenine dinucleotide (phosphate) [NAD(P)H] from their oxide form $NAD(P)^+$ is essential to the application of bioreduction in vitro[16–19]. Over the past 60 years, enzymatic[20,21], chemical[22], homogeneous[23–25], electro/photo-[26–30], and heterogeneous[31–33] approaches have been developed for NAD(P)H regeneration. In addition to enzymes, the only artificial catalysts with >95% NAD(P)H yield are the homogenous $[(arene)X(N, N')L]^{n+}$ (X = Rh, Ru, and Ir, L = ligand) complexes[34]. The high efficiency of these metal complexes is related with the formation of a hydride complex (M-H) as a hydrogen-carrier to deliver hydride to $NAD(P)^+$ [35], selectively yielding enzymatically active 1,4-NAD(P)H. The M-H species could be generated either with chemical reductants (formate[24], alcohols[36], or phosphite[37]) or with electron-coupled proton transfer through electro/photocatalysis[38,39].

[1]State Key Laboratory of Catalysis, Dalian Institute of Chemical Physics, Chinese Academy of Sciences, 457 Zhongshan Road, Dalian 116023, China. [2]University of Chinese Academy of Sciences, Beijing 100049, China. [3]Key Laboratory of the Ministry of Education for Advanced Catalysis Materials, Zhejiang Key Laboratory for Reactive Chemistry on Solid Surfaces, Institute of Physical Chemistry, Zhejiang Normal University, Jinhua 321004, China. ✉e-mail: yangqh@dicp.ac.cn

Alternatively, the NAD(P)H regeneration with $H_2$ as reductant would be more suitable for practical applications in terms of economy and scale-up[16]. Actually, the $H_2$-driven NAD(P)H regeneration has been explored using M[23,40] or supported metal NPs[31,33] as catalysts. However, the M usually gives lower activity, which is probably related to their lower $H_2$ dissociation ability[12]. The supported metal NPs suffer from the poor NAD(P)H selectivity arising from the non-oriented adsorption modes of NAD(P)$^+$ on the active sites[31]. Previously, the supported metal NPs and M were used separately in NAD(P)$^+$ hydrogenation, and their coupling catalysis remains unexplored. One of the reasons that hinder the coupling of metal NPs and M in one system may be the intrinsic cognition of the different hydrogenation mechanisms. For example, $H_2$ is homolytically cleaved to H on metal NPs in most cases[1,41], while it is dissociated to H$^-$/H$^+$ pair over M along with the formation of M-H[23].

In this work, we report the synergy of supported metal NPs and homogeneous M to boost NAD(P)H regeneration using $H_2$. The coupling of [Cp*Rh(bpy)($H_2O$)]$^{2+}$ (**1**, bpy = 2,2′-bipyridine) and supported Ru NPs simultaneously enhanced both activity and selectivity for NAD(P)$^+$ hydrogenation, suggesting the successful synergy of the strong $H_2$ dissociation ability of supported Ru NPs and the steric effect of M in one reaction. The hydrogenation mechanism for the synergy system was elucidated, and the synergy effect is proved to be a general phenomenon that can be extended to other supported metal NPs (Rh, Pd, Pt, and Ni) and M (Ru, Rh, Ir).

## Results

### Synergy of homogeneous and heterogeneous catalysts

Previously, our group reported that Pt/TiO$_2$ with 63.4% NADH selectivity is a relatively selective catalyst for NAD$^+$ hydrogenation, which may be related with the preferential adsorption of the C = O of NAD$^+$ on TiO$_x$ in the vicinity of Pt induced by strong metal-support interaction (SMSI)[31]. The formation of side products in metal NPs catalyzed NAD$^+$ hydrogenation may be caused by the NAD$^+$ adsorption on facet metal atoms by a parallel configuration. Our previous report showed that TP (1,3,5-triformylphloroglucinol)-TTA (4,4′,4″-(1,3,5-triazine-2,4,6-triyl) trianiline) polymer could efficiently cap the metal atoms in facet[42]. Herein, X/TiO$_2$ catalysts were chosen as catalysts and coated with a β-ketoenamine linked TP-TTA polymer layer formed by the condensation of TP and TTA (Fig. 1a). First, Ru/TiO$_2$@10TP-TTA (Fig. 1a) was tested in NAD$^+$ hydrogenation. A pH of 8.7 was chosen for NAD$^+$ hydrogenation according to previous report that the basic condition was beneficial for higher NADH selectivity using $H_2$ as reductant[31]. The

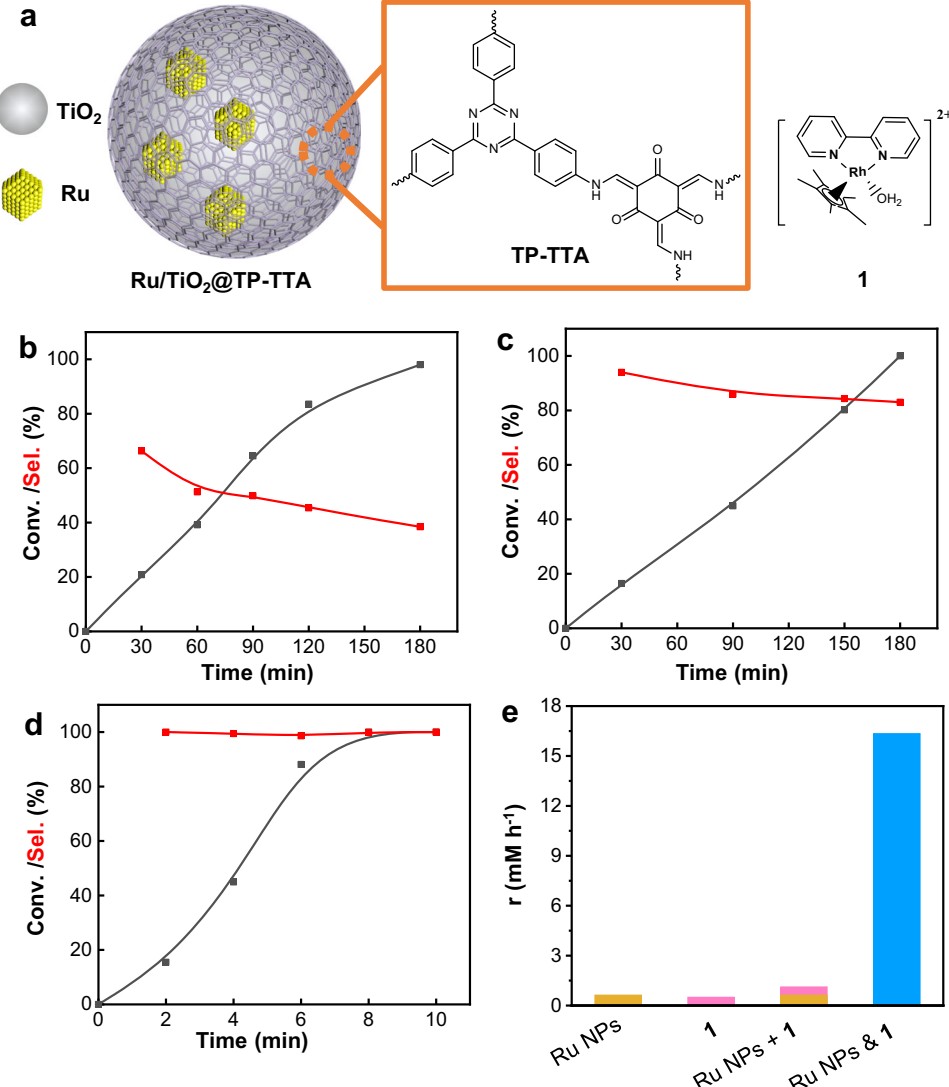

**Fig. 1 | Structures and hydrogenation performance of Ru/TiO$_2$@TP-TTA and 1.**
**a** Schematic illustration of the structures of Ru/TiO$_2$@TP-TTA and **1**. Reaction profiles of NAD$^+$ hydrogenation catalyzed by **b** Ru/TiO$_2$@10TP-TTA, **c 1** and **d** Ru/TiO$_2$@10TP-TTA & **1**. **e** Conversion rate (r) of NAD$^+$ hydrogenation of different catalysts, r was calculated with the linear part in the kinetic curve. Reaction conditions: 0.29 μmol Ru and/or 0.067 μmol Rh, 1.5 mM NAD$^+$, 2 mL of 0.1 M phosphate buffer (PB, pH = 8.7), 37 °C and 2 MPa $H_2$.

NAD$^+$ conversion and NADH selectivity were determined through an established enzymatic analysis method (see Methods). At 37 °C and 2 MPa H$_2$, Ru/TiO$_2$@10TP-TTA alone could catalyze NAD$^+$ hydrogenation to NADH, but the NADH selectivity decreased gradually from 66.3% to 38.4% along with the NAD$^+$ conversion increasing from 20.8% to 98.0% (Fig. 1b). The low selectivity is a general problem for NADH regeneration over supported metal catalysts[31]. Further, **1**, the most widely employed organometallic complex in NADH regeneration, was tested in NAD$^+$ hydrogenation under similar conditions. It needs as long as 180 min to reach full conversion, and the selectivity gradually decreased to 83.0% with the reaction time (Fig. 1c). After coupling Ru/TiO$_2$@10TP-TTA with **1**, >99% selectivity was achieved at full conversion within 10 min (Fig. 1d). Furthermore, the conversion rate (r) for Ru/TiO$_2$@10TP-TTA & **1** was determined to be 16.34 mM h$^{-1}$, about 15-fold to the sum total of the conversion rates of **1** and Ru/TiO$_2$@10TP-TTA used separately (Fig. 1e). Therefore, the catalytic activity was increased by an order of magnitude by coupling of Ru/TiO$_2$@10TP-TTA and **1**. The specific activity (normalized to all of the metals) was calculated to be 91.7 mol$_{NAD^+}$ mol$^{-1}_{metal}$ h$^{-1}$ for Ru/TiO$_2$@10TP-TTA & **1**. Ru/TiO$_2$@10TP-TTA & **1** is more selective than the reported homogeneous and heterogeneous catalysts (Supplementary Table 1). These results signify the advantage of the synergy of heterogeneous and homogenous catalysts in NADH regeneration. As far as we know, the cooperation of heterogeneous and homogenous catalysts has been rarely reported for selective hydrogenation.

To further elucidate the synergy of Ru NPs and **1**, the influence of Rh/Ru ratio on the catalytic performance was also investigated (Fig. 2a). The selectivity sharply increased from 75.6% to >99% by increasing the Rh/Ru ratio from 0.05 to 0.09, demonstrating that a small amount of **1** can significantly improve the selectivity. When the Rh/Ru ratio further increased, NADH selectivity remained at >99%. The conversion of NAD$^+$ was increased continuously along with the increment of Rh/Ru ratio, which suggests that the NAD$^+$ hydrogenation rate was possibly dominated by the formation rate of Rh-H species.

The catalytic performance of the coupling system of Ru/TiO$_2$@10TP-TTA and **1** was also tested in more challenging conditions. Even under neutral conditions, the coupling system also showed 100% conversion with >99% selectivity in NAD$^+$ hydrogenation (Fig. 2b), suggesting its good pH adaptability. The coupling system was further tested at 25 °C and 1 bar H$_2$. Again, NAD$^+$ could be fully converted to NADH with >99% selectivity (Fig. 2c). Very promisingly, NADPH, the phosphorylated form of NADH, was also successfully regenerated with >95% yield using the coupling system (Fig. 2d). Therefore, the coupling system presents excellent environmental tolerance properties.

### Hydride relay from heterogeneous to homogeneous catalyst

**1** was generally used as a mediator in chemical reduction of NAD$^+$ or in photocatalytic/electrocatalytic NAD$^+$ reduction. Using H$_2$ as reductant, **1** should be responsible for both H$_2$ dissociation and hydride transfer. The low H$_2$ dissociation activity of the positively charged Rh ion may cause the lower activity of **1** in NAD$^+$ hydrogenation. The electron-rich metal NPs facilitate the activation of H$_2$, but the non-oriented adsorption of NAD$^+$ on Ru NPs leads to the low selectivity of NADH. The significant improvement in both the activity and selectivity by coupling Ru/TiO$_2$@10TP-TTA and **1** suggests that Ru NPs is responsible for H$_2$ dissociation and **1** governs the NAD$^+$ reduction using the activated hydrogen species. A H$^*$ relay was possibly involved in NAD$^+$ hydrogenation with the coupling catalytic system.

To confirm that **1** acts as a homogeneous H$^*$ acceptor in NAD$^+$ hydrogenation catalyzed by Ru/TiO$_2$ & **1**, Ru/TiO$_2$ was filtrated out after NAD$^+$ conversion was above 20%. Then, HCOONa was added into the filtrate to further drive the reaction. The subsequent reaction profile

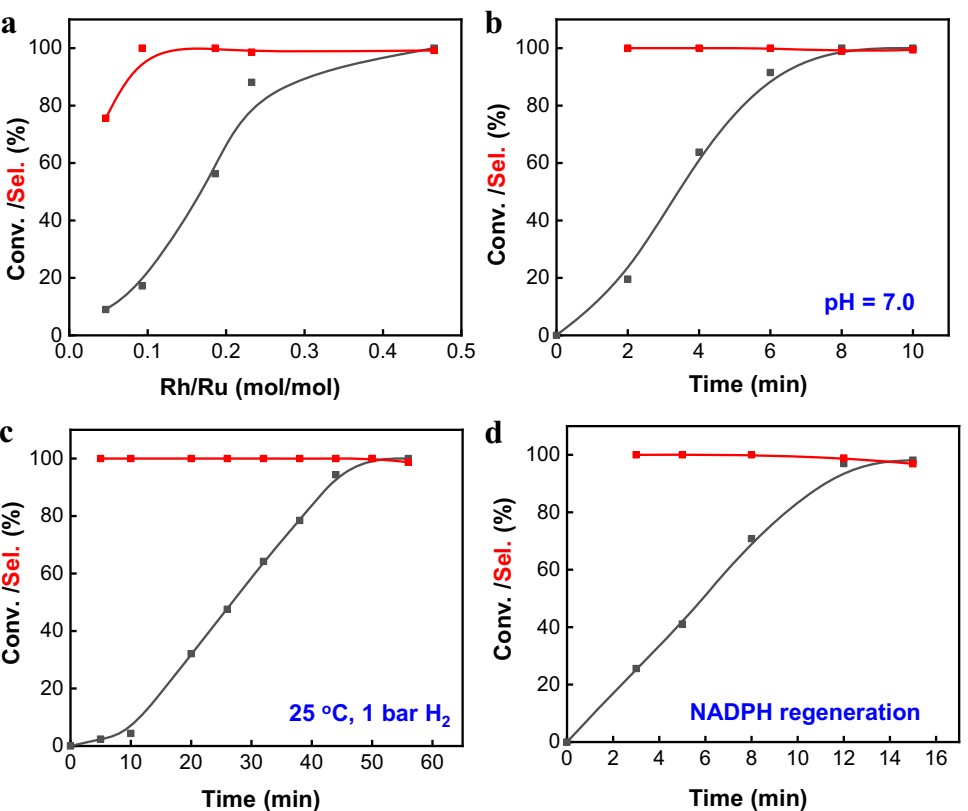

**Fig. 2 | Catalytic performance of Ru/TiO$_2$@10TP-TTA & 1.** NAD$^+$ hydrogenation (**a**) with different Rh/Ru molar ratio with a fixed Ru amount of 0.29 μmol, 6 min, **b**, **c** under varied conditions, (**d**) in NADP$^+$ hydrogenation. The reaction conditions were similar to those in Fig. 1 and the changed conditions were marked in the Figures.

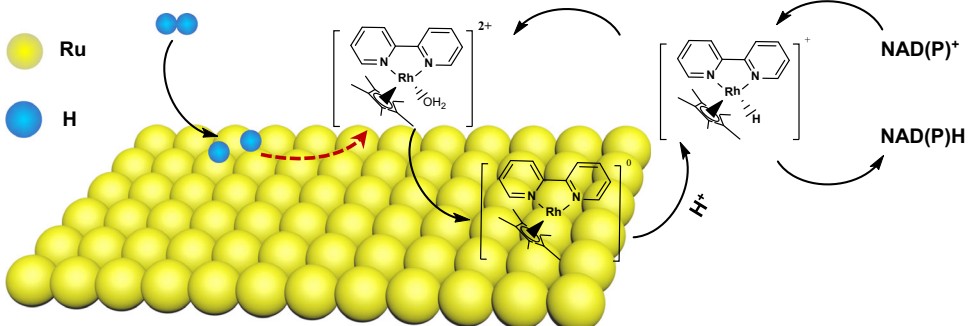

**Fig. 3 | Mechanism illustration.** Plausible mechanism of NAD(P)⁺ hydrogenation for the homogenous-heterogeneous coupling catalyst.

after the addition of HCOONa was well matched with that of the normal hydride transfer reaction for **1** using HCOONa as hydrogen source (Supplementary Fig. 1), indicating that **1** was well maintained in the hydrogenation process. The detection of **1** in the reaction system with spectroscopy techniques (e.g., UV-vis and NMR) was not successful due to the interference of NAD⁺/NADH.

Generally, Rh-H species is formed in the transfer hydrogenation of NAD⁺ using formate as reductant. Indeed, the appearance of signal at −11.5 ppm in the ¹H NMR spectrum of **1** after the addition of HCOONa (Supplementary Fig. 2a) confirmed the formation of Rh-H species. However, no signals could be observed in the negative region in the ¹H NMR spectrum of Ru/TiO₂@10TP-TTA & **1** after H₂ treatment (Supplementary Fig. 2b). Actually, after the treatment with H₂, the color of Ru/TiO₂ tuned from grey to dark-purple whereas the solution of **1** maintained a yellow color (Supplementary Fig. 3), which is different from the homogeneous dark-purple solution in the formate-driven system. These results suggested that the reduced Rh species was deposited on the surface of Ru/TiO₂ rather than being free in the solution. This explains the absence of Rh-H signal in the ¹H NMR spectrum. Previously, it has been demonstrated that the reduced [Cp*Rh(I)(bpy)]⁰ was deposited on the surface of electrode in the electrocatalytic reduction of **1** under alkaline conditions because of the sluggish proton coordination to metal center[35]. Herein, the electronic state of the Rh species deposited on Ru/TiO₂ (obtained by the H₂ treatment of Ru/TiO₂ in the presence of **1**) was characterized by X-ray photoelectron spectroscopy (XPS) technique (Supplementary Fig. 4). Two peaks at 308.3 eV and 312.9 eV could be identified in the Rh 3*d* spectra, respectively corresponding to the 3*d*₅/₂ and 3*d*₃/₂ binding energy (BE) of Rh. The BE of Rh 3*d*₅/₂ of the Rh species deposited on Ru/TiO₂ was between the fresh **1** (309.0 eV)[27,38] and the metallic Rh (ca. 307.0 eV)[43,44], but similar to that of Rh-H species (308.4 eV) isolated from **1** treated with HCOONa. With these combined results, it could be proposed that [Cp*Rh(bpy)H]⁺ is generated on the surface of Ru/TiO₂ after the incorporation of proton.

Isotopic experiments were performed to shed some light on the H* relay between the Ru NPs and **1** by using D₂O/H₂ and H₂O/D₂. As shown in the ¹H NMR spectrum of NADH (Supplementary Fig. 5), the signals between 2.6–2.8 ppm are assigned to the characteristic peaks of C4 hydrogen (H_{C4}), while the peak in 6.9 ppm belongs to the C2 hydrogen (H_{C2})[45,46]. The control experiment shows no H/D exchange between NADH and D₂O in the presence of Ru/TiO₂@10TP-TTA & **1**. Then, the NAD⁺ hydrogenation was carried out with D₂O/H₂ using NAD⁺ concentration as high as 15 mM. In this case, a slightly lower selectivity of 92.0% was obtained at full NAD⁺ conversion through enzymatic analysis, which was consistent well with that determined via ¹H NMR (91%, the byproduct is 1,6-NADH) (Supplementary Fig. 6), suggesting that a desirable activity and selectivity could still be maintained at high NAD⁺ concentration. The ¹H NMR spectrum of the obtained NADH shows H_{C2}/H_{C4} ratio of ca. 1:1.5, suggesting that H_{C4} in NADH was partially deuterium-labeled. Besides, a weak signal at 7.1 ppm assigned to an enzymatically inactive product 1,6-NADH[47] was

observed, probably caused by the high NAD⁺ concentration used herein. By using H₂O/D₂, the H_{C2}/H_{C4} ratio of ca. 1:1.5 was achieved once again for NADH (Supplementary Fig. 7). The presence of deuterium-labeled NADH in D₂O/H₂ suggests the participation of H₂O in NAD⁺ hydrogenation. Therefore, these results demonstrated that the proton transferred to NAD⁺ was originated from both H₂ and H₂O.

Kinetic isotope experiments were further carried out and a typical secondary kinetic isotope effect (KIE) value of 1.3 was obtained with H₂/D₂ (Supplementary Fig. 8). However, a more pronounced KIE value (2.1) was observed by switching the solvent from H₂O to D₂O, suggesting that the H₂O was involved in the rate-determining step. On the basis of the above results, it could be speculated that the H* relay from the Ru NPs to **1** was assisted with H₂O.

With the aforementioned results, the following pathway may be involved in the hydrogenation of NAD⁺ with the homogenous-heterogeneous coupling system (Fig. 3). H₂ was first dissociated into H* over Ru NPs. Subsequently, **1** was reduced to [Cp*Rh(I)(bpy)]⁰ using H*, and the reduced Rh species was deposited on the surface of Ru NPs. Rh-H species was then formed after the coordination of proton to the metal center. Notably, the proton could be originated from both H₂O and the concomitant proton during the reduction of **1**. Ultimately, the hydride was transferred to NAD(P)⁺ from [Cp*Rh(bpy)H]⁺ through a ring-slipped mechanism[48], meanwhile **1** was re-dissolved in the solution for the next catalysis cycle.

## Competitive reaction routes in the coupling system

Ru/TiO₂ and Ru/TiO₂@5TP-TTA were also prepared in a similar method to Ru/TiO₂@10TP-TTA, but with different amounts of TP-TTA. Thermogravimetric analysis (TGA, Supplementary Table 2 and Supplementary Fig. 9) showed that the organic content was 3.0 w.t.% and 7.1 w.t.% for Ru/TiO₂@5TP-TTA and Ru/TiO₂@10TP-TTA, respectively. The BET surface area increased gradually along with the increment of the organic content (Supplementary Fig. 10 and Supplementary Table 2) because of the porous structure of TP-TTA[49]. The results of transmission electron microscopy (TEM), high angle annular dark field-scanning transmission electron microscopy (HAADF-STEM), and Fourier transform infrared (FT-IR) spectroscopy (Fig. 4 and Supplementary Figs. 11–13) confirmed the existence of 1.8 nm and 3.6 nm TP-TTA layer on Ru/TiO₂@5TP-TTA and Ru/TiO₂@10TP-TTA, respectively. The TEM images (Supplementary Figs. 11, 12) showed that the average particle size of Ru was in the range of 1.5-1.9 nm for the three catalysts. The energy-dispersive X-ray (EDX) spectroscopy results of Ru/TiO₂@10TP-TTA further revealed the uniform encapsulation of TP-TTA over TiO₂ and the close contact of Ru NPs with TP-TTA (Fig. 4c). The dispersion of Ru analyzed by CO chemisorption showed that the dispersion of Ru was 12.2%, 6.7% and 2.9% for Ru/TiO₂, Ru/TiO₂@5TPTTA, and Ru/TiO₂@10TP-TTA, respectively, although these catalysts had similar Ru particle size (Supplementary Table 2). This suggests that the Ru NPs were covered by a polymer layer for Ru/TiO₂@TP-TTA, which is consistent with a recent report by our group[42].

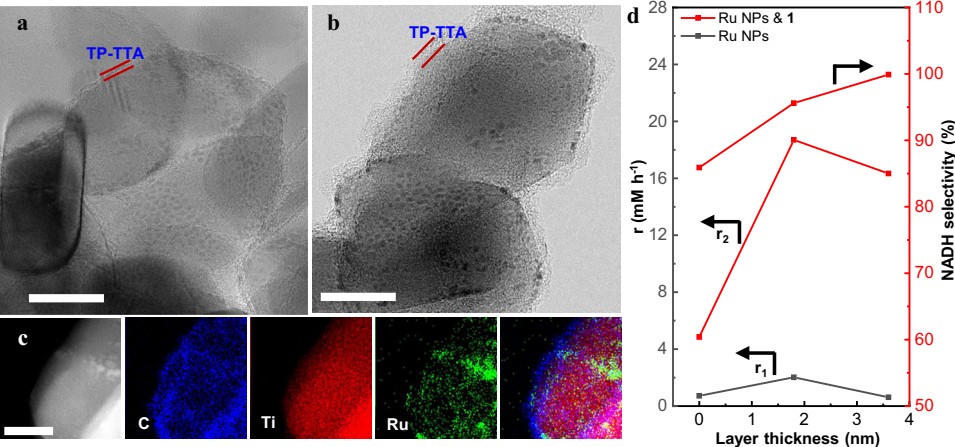

**Fig. 4 | Characterizations and hydrogenation performance of Ru/TiO$_2$@TP-TTA.** TEM images of **a** Ru/TiO$_2$@5TP-TTA, **b** Ru/TiO$_2$@10TP-TTA, **c** STEM image of Ru/TiO$_2$@10TP-TTA and the corresponding EDX mapping and the reconstructed overlay image (scale bar 20 nm). **d** The relationship between the layer thickness of TP-TTA and the conversion rate/NADH selectivity of Ru/TiO$_2$@TP-TTA & **1**.

The in situ FT-IR spectra of CO adsorption (Supplementary Fig. 14) further revealed the suppression of CO adsorption after the integration of TP-TTA with Ru/TiO$_2$ (details see Supplementary information). Furthermore, all of the adsorption peaks shifted to lower wavenumbers after the coating of TP-TTA, indicating the more electron-rich surface of Ru for Ru/TiO$_2$@TP-TTA.

After coupling Ru/TiO$_2$ or Ru/TiO$_2$@5TPTTA with **1**, both the activity and NADH selectivity are once more significantly enhanced in NAD$^+$ hydrogenation as compared with the corresponding single counterpart (Supplementary Table 3). This further confirms the synergy between M and metal NPs. However, the NADH selectivity strongly depends on the thickness of TP-TTA layers. The NADH selectivity of the coupling system increases from 85.9% to 95.6% and then to >99% as increasing the TP-TTA layer thickness of Ru/TiO$_2$@TP-TTA from 0 to 1.8 nm and then to 3.6 nm (Fig. 4d). This suggests that NAD$^+$ was also hydrogenated on the surface of Ru NPs for the former two samples. Thus, there are two competitive NAD$^+$ hydrogenation routes in the coupling catalyst, the selective one catalyzed by **1** and the non-selective one catalyzed by Ru NPs.

In the coupling system, the competitive reduction of **1** and NAD$^+$ on Ru NPs may determine the selectivity, which means that the high selectivity requires the fast hydrogenation rate of **1** on Ru NPs but a slow one of NAD$^+$. Although the net hydrogenation rate of **1** on Ru NPs could not be determined directly, it could be roughly estimated from the NAD$^+$ conversion rate in the coupling system. As shown in Fig. 3d, both NAD$^+$ conversion rates on Ru NPs ($r_1$) and in the coupling system ($r_2$) were enhanced for Ru/TiO$_2$@TP-TTA with 1.8 nm TP-TTA overlayer, which was probably originated from the electron-rich surface of Ru as demonstrated by the in situ FT-IR results. However, it should be pointed out that $r_2$ was more pronouncedly enhanced than $r_1$. For Ru/TiO$_2$@TP-TTA with 3.6 nm TP-TTA layer, $r_2$ decreased slightly, but a lowest $r_1$ was observed. This means that the selective NAD$^+$ hydrogenation route gradually dominants with the increase of the TP-TTA layer thickness, resulting in the enhanced selectivity.

The NAD$^+$ hydrogenation rate of Ru/TiO$_2$@TP-TTA & **1** shows a volcano curve with the TP-TTA layer thickness (Fig. 3d), which is the combined effect of the hydrogenation activity for NAD$^+$ and **1**. As compared with NAD$^+$, **1** has smaller molecular size. Considering the diffusion barrier, NAD$^+$ hydrogenation may be more severely suppressed on Ru/TiO$_2$@TP-TTA, providing more Ru sties for **1** hydrogenation. This explains the higher activity of Ru/TiO$_2$@10TP-TTA & **1** than Ru/TiO$_2$ & **1**. The lower activity of Ru/TiO$_2$@10TP-TTA & **1** than Ru/TiO$_2$@5TP-TTA & **1** is caused by the lower surface exposure degree of Ru/TiO$_2$@10TP-TTA.

Previously, it was surprising to find that **1** alone delivered gradually decreased selectivity using H$_2$ as the reductant (Fig. 1c). The Rh species in the solution after treatment with H$_2$ was separated by removing water and characterized by XPS technique. The deconvoluted spectrum of Rh 3$d$ clearly revealed a peak at 307.6 eV (Supplementary Fig. 15), which could be attributed to the metallic Rh species[50]. Thus, it's inferred that the decreased selectivity was possibly caused by the partially reduction of **1** to metallic Rh, leading to the non-selective hydrogenation of NAD$^+$. No obvious acceleration in the reaction rate together with 83.0% NADH selectivity at full conversion implies the low activity of the metallic Rh species formed under the operating conditions. As mentioned above, no metallic Ru species were observed on Ru/TiO$_2$ by the treatment of Ru/TiO$_2$ & **1** in H$_2$. This is possibly related with the fact that the reduced **1** only adhered on the surface of Ru/TiO$_2$ and the lack of the free mobility prevents the aggregation of Rh species to form metallic Rh.

## Generality of the cooperation of homogeneous and heterogeneous catalysis

Since metal NPs is responsible for H$_2$ dissociation in the coupling systems, we speculate that other metal NPs with strong H$_2$ activation ability may play an equivalent role. To verify that, Rh/TiO$_2$@5TP-TTA, Pd/TiO$_2$@5TP-TTA and Pt/TiO$_2$@5TP-TTA were also prepared using the same procedure as that of Ru/TiO$_2$@10TP-TTA. The particle size estimated from the TEM images (Supplementary Fig. 16) was 2.0, 2.3, and 1.6 nm for Rh, Pd, and Pt NPs, respectively. As shown in Supplementary Table 3, the NADH selectivity of Rh/TiO$_2$@5TP-TTA, Pd/TiO$_2$@5TP-TTA and Pt/TiO$_2$@5TP-TTA within 30 min was 15.3%, 51.5% and 67.8%, respectively, and the NAD$^+$ conversion was 97.8%, 87.2% and 100%, respectively. Considering that the NAD$^+$ conversion may influence the NADH selectivity, the reaction time of the coupling system for the (nearly) full conversion of NAD$^+$ was carefully chosen to make the reasonable selectivity comparison. As expected, after coupled with **1**, the greatly enhanced activity and selectivity were observed for all of the tested metal NPs. Moreover, we noticed that the improvement on the selectivity was not directly related with the selectivity of metal NPs. For example, Pt/TiO$_2$@5TP-TTA is more selective than other catalysts, but it shows the lowest selectivity after coupling with **1**. The selectivity of the coupling system shows a negative correlation with the activity of metal NPs for NAD$^+$ hydrogenation. This is caused by the competitive reduction of NAD$^+$ on **1** and on metal NPs. The NADH selectivity in the coupling system is the sum up of the non-selective NAD$^+$ hydrogenation on metal NPs and the selective NAD$^+$ hydrogenation on reduced **1**. To this end, it is reasonable to observe that higher NADH selectivity

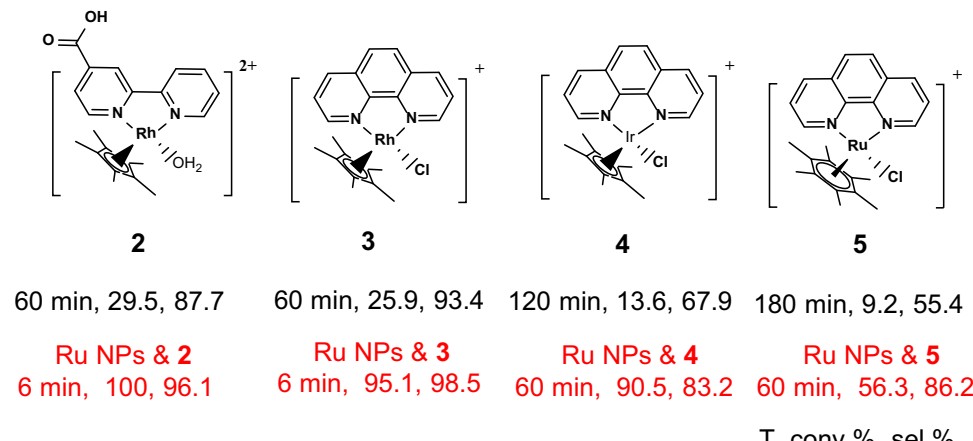

**Fig. 5 | Structures of 2-5 and their NAD+ hydrogenation performance.** Reaction conditions were the same as those used in Fig. 1 except that 0.25 μmol Ir and 0.60 μmol Ru complexes were used.

was obtained on metal NPs with less activity for NAD+ hydrogenation. Furthermore, X/TiO2@10TP-TTA (X = Rh, Pd, and Pt, Supplementary Fig. 17) was also prepared, and higher selectivity was observed for all samples in comparison with their corresponding 5% TP-TTA counterpart, which is in a similar tendency to that of Ru/TiO2@TP-TTA. Above results verified that metal NPs with lower NAD+ reduction activity greatly promoted the selectivity in the coupling system.

In comparison with noble metal, the non-precious metal catalysts are more attractive in view of not only their abundance but also the inhibition the non-selective NAD+ hydrogenation route because of their low hydrogenation activity in mild conditions. Ni/TiO2 (Supplementary Fig. 18) with 2 wt% Ni loading and Ni particle size of 14.6 nm was prepared by wet-impregnation method. Ni/TiO2 only provides a low conversion of 4.7% in 40 min in NAD+ hydrogenation. When coupled with **1**, significantly enhanced activity and selectivity were observed with 100% NAD+ conversion and 95.1% NADH selectivity. These results suggest the possibility of using non-precious metal catalysts as the candidates for practical applications.

As we discussed above, the role of **1** was to transfer hydride, therefore, **1** would not be the only candidate that could synergistically work with metal NPs, since a series of homogenous catalysts with a similar structure to **1** have been developed for NAD(P)H regeneration[34]. To verify this, **2-5** complexes were used to couple with Ru/TiO2@10TP-TTA (Fig. 5). The coupling of **2** (with carboxyl-substitution on bpy ligand) and Ru NPs showed a NADH yield of 96.1%, higher than that of **1** (88.1%) in NAD+ hydrogenation. **3** (with 1,10-phenanthroline as ligand) gave a 25.9% conversion with 93.4% selectivity in 60 min. After the introduction of Ru/TiO2@10TP-TTA, the conversion significantly enhanced to 95.1% along with the enhanced selectivity of 98.5% in 6 min. In addition to Rh complexes, the 1,10-phenanthroline complexes of Ir (**4**) and Ru (**5**) were also synthesized according to a reported method[25], and the synergy effect was also observed despite the lower activity compared with those of Rh complexes. The above results confirmed the universality of the synergy effect between metal NPs and homogenous complexes in NAD+ hydrogenation. Furthermore, the coupling in situ NADPH regeneration with enzyme for the chemoenzymatic reduction of acetophenone was demonstrated in Supplementary Tables 4, 5, and Supplementary Fig. 19.

In summary, an efficient catalytic system for NAD+/NADP+ hydrogenation was demonstrated by coupling metal complex with supported metal NPs. The activity was promoted to 15 times and the selectivity was increased to >99% in NAD+ hydrogenation by coupling of supported Ru NPs and [Cp*Rh(bpy)(H2O)]2+. The generality of the synergy effect was verified by the significantly enhanced activity and

selectivity in NAD+ hydrogenation by coupling different types of supported metal NPs (Ru, Rh, Pd, Pt, and Ni) and metal complexes (Ru, Rh, and Ir complexes). The mechanism investigation revealed that the synergy was a result of the combination of the strong H2 dissociation ability of metal NPs and the steric effect of M. The selective formation of enzymatically active 1,4-NAD(P)H is attributed to the efficient H* rely from metal NPs through M to NAD+. Furthermore, coupling in situ NADPH regeneration with enzyme was demonstrated. This work highlights the power of synergy between heterogeneous and homogenous catalysts in addressing the challenges facing homogeneous and heterogeneous catalysis.

## Methods
### Preparation of organometallic complexes
M complexes were synthesized according to the reported methods[25,51]. For the synthesis of **1** and **2**, (Cp*RhCl2)2 was dispersed in methanol, the mixture turned to a clear orange solution after two equivalents of 2,2′-bipyridine or 2,2′-bipyridine-4-carboxylic acid were added. Precipitate was formed after the addition of diethyl ether into the solution, which was dissolved in water for further use. **3**, **4**, and **5** were synthesized through the reaction of the corresponding [(arene)MCl2]2 salt with 1,10-phenanthroline with stoichiometric ratio in dichloromethane at room temperature. After the evaporation of dichloromethane, the obtained samples were dissolved in water.

### Preparation of X/TiO2@xTP-TTA
Ru/TiO2@xTP-TTA (x refers to w.t.% of TP-TTA) was prepared according to a reported method[52] with slight modification. In a typical procedure, TiO2 (300 mg) was dispersed in an aqueous solution of RuCl3 (3 mL, 3 mg Ru). After stirring overnight, the slurry was dried through rotary evaporation followed by calcination in static air for 4 h at 400 °C. Next, the resultant sample with 1 wt% Ru loading was well dispersed in 20 mL dioxane using sonication. Then, the desired amounts of TP and TTA with stoichiometric ratio were separately dissolved in dioxane (1 mL) and dropped into the suspension under stirring. The mixture was stirred at room temperature under N2 for 4 h and further maintained at 100 °C for another 20 h. After cooling down to room temperature, the sample was collected by centrifugation and washed with ethanol. Finally, the sample was reduced under H2 at 200 °C for 2 h. The same procedure was employed for the preparation of Rh, Pd, and Pt/TiO2@xTP-TTA with metal loading of 1% using RhCl3·xH2O, Na2PdCl4, and H2PtCl6 as the precursor, respectively. Ru/TiO2 was synthesized using the same procedure without the integration of TP-TTA.

For the preparation of $Ni/TiO_2$ with 2 wt% Ni loading, $TiO_2$ (300 mg) was added in an aqueous solution of $Ni(NO_3)_2·6H_2O$ (3 mL, 6 mg Ni) and stirred at room for 1 h. The temperature was then increased to 50 °C to dry the slurry, then it was dried in an oven at 100 °C for 2 h and calcinated for 3 h at 550 °C. The sample was reduced under hydrogen flow for 3 h at 600 °C. This catalyst was denoted as $Ni/TiO_2$.

## General procedure for NAD$^+$ hydrogenation

NAD$^+$ hydrogenation was carried out in a stainless-steel autoclave (300 mL) or double-necked flask (25 mL). In a typical experiment, heterogenous catalyst (0.29 μmol metal) and/or homogenous catalyst (0.067 μmol metal) and phosphate buffer (PB, 2 mL, pH = 8.7) containing NAD$^+$ (1.5 mM) were loaded into ampule tube. The system was purged with $H_2$ for 6 times to remove the air, then the reaction was carried at 37 °C, 2 MPa $H_2$. After the reaction, the catalyst was filtrated out and 0.5 mL filtrate was taken out and diluted to 10 mL by PB (pH = 7.0) for further analysis.

## Analysis of NAD(P)$^+$ and NAD(P)H

NAD$^+$ and NADH were analyzed quantitatively through an established enzymatic method developed by our group[31]. Generally, the above-diluted solution was firstly monitored by UV-vis spectrometer. Afterwards, 5 mL solution was incubated with 10 μL alcohol dehydrogenase (ADH) and 10 μL propionaldehyde for 5 min, another 5 mL solution was incubated with 2.5 U glucose dehydrogenase (GDH) and 2.25 mg glucose for 10 min. After incubation, the solutions were again monitored by UV-vis spectrometer. NADH yield and NAD$^+$ conversion could be determined through the difference between the absorbance at 340 nm before and after incubation with ADH and GDH, respectively.

NADP$^+$ conversion was determined using the same method as that of NAD$^+$. The NADPH yield was analyzed through the incubation with aldehyde ketone reductase (AKR, 2 μL) and acetone (10 μL) for 5 min.

## Data availability

All data generated in this study are provided in the Supplementary Information/Source Data file. Source data are provided with this paper.

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

## Acknowledgements

This work was supported by the National Natural Science Foundation of China (21733009, 21972134).

## Author contributions

M.W. did most of the experiments and wrote the original manuscript; Z.Z. carried out NMR characterization and revised the manuscript; C.L. helped with the characterization and analysis; H.L. and J.L. synthesized **1**; Q.Y. designed and supervised the project. The manuscript was written through the contributions of all authors.

## Competing interests

The authors declare no competing interests.
