## [Peer Review File · Nature Communications]

Title: Synergy of metal nanoparticles and organometallic complex in NAD(P)H regeneration via relay hydrogenationREVIEWER COMMENTS

Reviewer #1 (Remarks to the Author):

This manuscript reported a combined heterogeneous and homogeneous catalytic system for NADH regeneration. It does show some encouraging results (100% conversion in 10 min with ~100% selectivity). However, the manuscript was poorly prepared with little rationale on their choice of catalysts combination. The Discussion section is rather inadequate. It should be a conclusion. Moreover, some key references were not clearly presented (e.g. #48), which made readers finding it difficult to follow. For a prestige journal like Nature Communication, the manuscript needs to be of the highest quality. At the current status, this work does not meet the standard for publication.

Major comments:

1. The whole manuscript gives little reasoning on choices of catalyst design as well as reaction parameters. For example, the choice of support for Ru, which is a known hydrogenation catalyst but not the most common, was not clearly explained. TiO₂ again is not the most common choice for standard hydrogenation. More intriguingly, the function of the TP-TTA coating was also not explained at all. When I read the manuscript, they are all like some mystery.
2. Generally, such catalysis work should compare on conversion rate, i.e. basic kinetic analysis, not 100% conversion at a certain time point. Here, the rate should be given as per mole of catalyst, making it easier for readers to compare with data from literature.
3. The whole manuscript was poorly prepared. Discussion is not a discussion. Results section starts with a figure, so as one subsection in R+D. Notations, notably TP and TTA, were not even explained in the whole manuscript, only mentioned in SI. Some references were not put properly (e.g. no volume, no journal and/or no page number).
4. The initial characterisation of the heterogeneous catalysts (Ru/TiO₂@10TP-TTA) use x=10 but the final sections, which is the most important section in my opinion, all other catalysts (e.g. Rh, Pt and Pd) were prepared with x=5. What if these catalysts perform much better with x=10?
5. Also in the combined reaction for acetophenone, the production rate is rather low, 20 μmol in 15 h. Also, the function of Ni/TiO₂ was not clear, except being used to "suppress background reaction". What is this background reaction? Why it makes such a difference in the yield and TON? Also, this part does not seem to involve the Ru/TiO₂@TP-TTA catalyst. I thought the whole aim of this work is about the combined "M" and Ru/TiO₂@TP-TTA system. This section seems to be detached from the rest of the work.

Reviewer #2 (Remarks to the Author):

Yang and coworkers report a novel method for the hydrogenation/regeneration of NAD (P)⁺ to NAD(P)H with H₂ catalyzed by a tandem of supported metal nanoparticles (NPs, based on Ru, Rh, Pt, Pd and Ni) and an organometallic complex (M, based on Rh, Ir and Ru). The authors observe that the presence in the reaction media of both heterogeneous and homogeneous catalysts greatly improves the catalytic performance with selectivity up to > 99% and reaction rates about one order of magnitude higher than the corresponding activity for the metal NPs and M alone.

In addition, the tandem Ni/TiO₂ and M1 was also coupled to an enzyme (AKR) for the asymmetric hydrogenation of acetophenone reaching up to > 99% ee and at least 24 cycles of NADPH regeneration. The work is interesting and competently executed. For these reasons, I recommend the publication of the paper in Nat. Commun. once the following minor revisions will be addressed.

- 1) I would suggest to the authors to determine the Turnover Frequency of the catalyst instead the reaction rate to better compare the activity of the studied complexes with the literature benchmarks.
- 2) The explanation of the reaction mechanism showed in Scheme 1 (page 11) is not clear and hard to read. I would suggest to the authors to improve that part of the manuscript.
- 3) Do authors find any sign of speciation of NAD⁺ and NADH during their NMR isotopic studies (page 10 from line 10 on), as previously observed (see ACS Catal. 2017, 7, 7788–7796; this paper should be quoted)?
- 4) Characterization data (such as dimensions and distribution of the NPs) of Ni@TiO₂ are not given. Please, provide it.
- 5) The NMR spectra provided in the SI (in particular Figure S2 and S3) are unphased and of low quality. Please, improve the quality of the NMR spectra reported.
- 6) I would suggest to better elucidate the dependence of catalytic activity and selectivity on the thickness of TP-TTA and how it affects the formation of metallic Rh species.
- 7) I would suggest obtaining more insights on why a negative correlation between the activity of metal NPs and the selectivity of the coupling system for Rh/TiO₂@5TP-TTA, Pd/TiO₂@5TP-TTA and Pt/TiO₂@5TP-TTA is observed.
- 8) Enzymatic concentration is not given. Please, provide it.
- 9) Typos errors: page 6 a > 99%; SI fig. S6 H₂ not N₂.
- 10) The English of the manuscript is poor in many places making the meaning of some sentences little clear. For examples: 1) page 3, lines 12-13: "... it may help break through the plateau in catalysis"; 2) page 9, line 2: "... filtration reaction ..."; 3) page 10, line 9: "... the incorporation of proton into the ligand sphere ..." what does it mean?

Responses to Reviewer 1

This manuscript reported a combined heterogeneous and homogeneous catalytic system for NADH regeneration. It does show some encouraging results (100% conversion in 10 min with ~100% selectivity). However, the manuscript was poorly prepared with little rationale on their choice of catalysts combination. The Discussion section is rather inadequate. It should be a conclusion. Moreover, some key references were not clearly presented (e.g. #48), which made readers finding it difficult to follow. For a prestige journal like Nature Communication, the manuscript needs to be of the highest quality. At the current status, this work does not meet the standard for publication.

Response: We sincerely appreciate your careful review and valuable suggestions. It really helps us a lot to improve our work. The reasons for the choice of the catalysts was clarified in the revised manuscript. Discussions could be found in the section of Results. We have tried our best to meet the high quality of Nature Communication.

The questions are answered point-by-point as follows:

Major comments:

Question 1. The whole manuscript gives little reasoning on choices of catalyst design as well as reaction parameters. For example, the choice of support for Ru, which is a known hydrogenation catalyst but not the most common, was not clearly explained. TiO₂ again is not the most common choice for standard hydrogenation. More intriguingly, the function of the TP-TTA coating was also not explained at all. When I read the manuscript, they are all like some mystery.

Response: According to your suggestion, the reasons for the choice of the catalysts and the function of TP-TTA coating have been clarified in the revised manuscript in page 4 as follows “Previously, our group reported that Pt/TiO₂ with 63.4% NADH selectivity is a relatively selective catalyst for NAD⁺ hydrogenation, which may be related with the preferential adsorption of the C=O of NAD⁺ on TiO_x in the vicinity of Pt induced by strong metal-support interaction (SMSI)³¹. Therefore, TiO₂ was used as support for metal NPs in this work. To further increase NADH selectivity, the continuous metal sites which may cause unselective NAD⁺ adsorption should be avoided. To this end, M/TiO₂ catalysts were further coated with a β-ketoenamine linked TP-TTA polymer layer formed by the condensation of TP (1,3,5-triformylphloroglucinol) and TTA (4,4',4''-(1,3,5-triazine-2,4,6-triyl)trianiline) (Fig. 1a), considering that TP-TTA polymer could efficiently cap the continuous Pt sites based on our previous report⁴²”.

Question 2. Generally, such catalysis work should compare on conversion rate, i.e. basic kinetic analysis, not 100% conversion at a certain time point. Here, the rate should be given as per mole of catalyst, making it easier for readers to compare with data from literature.

Response: Thank you for your valuable suggestion. We calculated the specific activity in the kinetic region (normalized to all of the metals), and the results in this work together with the reported results were listed in Supplementary Table 1. To make it more readable, we added the specific activity in the revised manuscript in page 5.

Supplementary Table 1. Comparison of the results of NAD⁺ hydrogenation in this work and in literatures.

Catalyst	NAD ⁺ (mM)	T (°C)	H ₂ (atm)	pH	Sel. (%)	Specific activity (mol _{NAD⁺} mol ⁻¹ _{metal} h ⁻¹) ^a	Ref.
Ru/TiO ₂ @10TP-TTA & M1	1.5	37	20	8.7	>99	91.6	This work
	1.5	37	20	7.0	>99	90.7	
	1.5	37	1	8.7	>99	38.3	
	1.5	25	1	8.7	>99	13.6	
	15	37	20	9.4	92.0	47.2	
Pt/ana-450	1.5	37	1	8.7	63.4	19.9	1
Pt/SiO ₂	1.5	22	9	7.0	30	~512	2
Pt90Sn10/SiO ₂	0.1	22	9	7.0	~90	46.3	
[Ir(Cp [*])(4-(1H-pyrazol-1-yl-κN ²)benzoic acid-κC ³)(H ₂ O)] ₂ SO ₄	0.77	25	1	6.5	97 ^b	36	3
	0.77	25	1	7.0	97 ^b	~30	
	1.9	25	1	7.0	97 ^b	<10	

^aThe specific activity was calculated based on all of the metals used in reactions. ^bYield of NADH.

Question 3. The whole manuscript was poorly prepared. Discussion is not a discussion. Results section starts with a figure, so as one subsection in R+D. Notations, notably TP and TTA, were not even explained in the whole manuscript, only mentioned in SI. Some references were not put properly (e.g. no volume, no journal and/or no page number).

Response: Thanks for your kind reminding. We have carefully revised the manuscript. The discussion section was merged in the section of Results. TP and TTA were explained in the revised manuscript, and the improperly cited referenced were corrected.

Question 4. The initial characterisation of the heterogeneous catalysts (Ru/TiO₂@10TP-TTA) use x=10 but the final sections, which is the most important section in my opinion, all other catalysts (e.g. Rh, Pt and Pd) were prepared with x=5. What if these catalysts perform much better with x=10?

Response: Thank you for your valuable suggestion. As you suggested, M/TiO₂@10TP-TTA (M = Rh, Pd and Pt) was prepared and tested in NAD⁺ hydrogenation and the results were listed in Supplementary Table 3. Similar to Ru/TiO₂@xTP-TTA, M/TiO₂@10TP-TTA (M = Rh, Pd and Pt) catalysts are more selective than M/TiO₂@5TP-TTA (M = Rh, Pd and Pt). However, Ru/TiO₂@10TP-TTA still presents the best catalytic performance after coupling with **M1**.

Supplementary Table 3. Catalytic performance of NPs and **M1** in NAD⁺ hydrogenation^a.

Entry	Catalyst	Time (min)	Conv. (%)	Sel. (%)
1	M1	30	16.5	94.0
2	Ru/TiO ₂	30	23.9	48.5
3	Ru/TiO ₂ & M1	20	100	85.9
4	Ru/TiO ₂ @5TP-TTA	30	67.4	45.9
5	Ru/TiO ₂ @5TP-TTA & M1	5	100	95.6
6	Ru/TiO ₂ @10TP-TTA	30	20.8	66.3
7	Ru/TiO ₂ @10TP-TTA & M1	8	100	>99
8	Rh/TiO ₂ @5TP-TTA	30	97.8	15.3
9	Rh/TiO ₂ @5TP-TTA & M1	6	100	87.6
10	Rh/TiO₂@10TP-TTA & M1	6	100	90.3
11	Pd/TiO ₂ @5TP-TTA	30	87.2	51.5
12	Pd/TiO ₂ @5TP-TTA & M1	10	100	94.4
13	Pd/TiO₂@10TP-TTA & M1	30	97.8	96.8
14	Pt/TiO ₂ @5TP-TTA	30	100	67.8
15	Pt/TiO ₂ @5TP-TTA & M1	10	100	83.8
16	Pt/TiO₂@10TP-TTA & M1	10	100	87.9
17	Ni/TiO ₂	40	4.7	n.d.
18	Ni/TiO ₂ & M1	40	100	95.1

^aReaction conditions: 0.29 μmol NPs and/or 0.067 μmol Rh, 1.5 mM NAD⁺, 2 mL of 0.1 M phosphate buffer (PB, pH = 8.7), 37 °C and 2 MPa H₂.

Question 5. Also in the combined reaction for acetophenone, the production rate is rather low, 20 μmol in 15 h. Also, the function of Ni/TiO₂ was not clear, except being used to “suppress background reaction”. What is this background reaction? Why it makes such a difference in the yield and TON? Also, this part does not seem to involve the Ru/TiO₂@TP-TTA catalyst. I thought the whole aim of this work is about the combined “M” and Ru/TiO₂@TP-TTA system. This section seems to be detached from the rest of the work.

Response: Thank you for your question. The low yield of phenyl ethanol (76.6 μmol) is partly because of the small amount of NADP^+ (1.5 mM, 2 mL) used in the system. The efficiency of our catalytic system is superior to most of the reported artificial NAD(P)H regeneration systems, including transfer hydrogenation, photocatalysis and electrocatalysis, as shown in Supplementary Table 5. For example: $[(\eta^5\text{-C}_5\text{Me}_5)\text{Rh}(\text{phen})\text{Cl}]^+$ (Eur. J. Inorg. Chem. 2007, 4736-4742) is one of the most efficient homogenous catalysts for NAD(P)H regeneration, but the product yield and TON are much lower than our system.

Supplementary Table 5. Comparison of the results of chemoenzymatic reduction in this work and representative results in literatures.

NAD(P)H regeneration catalyst	Conditions	Time (h)	Amount of NAD(P) ⁺	Substrate	ee (%)	Product (μmol)	TON	Ref.
Ni/TiO ₂ & M2	37 °C, 2 MPa H ₂	15	1.5 mM, 2 mL	acetophenone	>99	76.6	24	This work
$[(\eta^5\text{-C}_5\text{Me}_5)\text{Rh}(\text{phen})\text{Cl}]^+$	37 °C, 0.1 M formate	24	1 mM, 1 mL	acetophenone	98	6.6	6.6	6
Rh@PMO	40 °C, 0.1 M formate	42	1.5 mM, 2 mL	4-phenyl-2-butanone	>98	17.2	5.7	7
CCGCMAQS P & M1	Photocatalysis	50	1 mM, 3 mL	acetophenone	>99	12.6	4.2	8
a-MoS _x	Electrocatalysis	4	1 mM, 2 mL	benzaldehyde	-	17.4	8.7	9
TP-COF & M1	Photocatalysis	0.2	2 mM, 3 mL	α -ketoglutarate	-	2.9	0.5	10

The background reaction means the hydrogenation of acetophenone over metal nanoparticles, which will greatly reduce the enantioselectivity of the catalytic system of enzyme coupled NADH regeneration. In this work, Ru/TiO₂@10TP-TTA was firstly chosen as a model catalyst to couple with enzyme because of its high NADH selectivity. However, it was found that Ru/TiO₂@TP-TTA could directly catalyze the hydrogenation of acetophenone, which results in the decrease in enantioselectivity. The similar phenomenon was observed for other supported noble metal catalysts (Supplementary Table 4.). In order to increase the enantioselectivity of enzyme coupled NADH regeneration system, Ni/TiO₂ was used for in situ NADH regeneration because Ni/TiO₂ is not active for acetophenone hydrogenation under current conditions.

Supplementary Table 4. Results of chemoenzymatic reduction of acetophenone.

NPs	Yield (μmol)	ee (%)
Ru/TiO ₂ @10TP-TTA	47.4	40.4
Rh/TiO ₂ @10TP-TTA	41.8	54.5
Pd/TiO ₂ @10TP-TTA	17.7	81.2
Pt/TiO ₂ @10TP-TTA	47.7	64.6
Ni/TiO ₂	76.6	>99

Reaction conditions: 0.29 μmol metal NPs (3.1 μmol Ni) and 0.268 μmol Rh (**M2**), 20 μL AKR, acetophenone (92 μmol), 3.0 μmol NADP⁺, 2 mL of 0.1 M PB, pH = 7.0, 37 °C and 2 MPa H₂, 15h.

The aim of our work is to demonstrate the synergy effect of supported metal NPs and homogeneous catalysts in the H₂-driven NADH regeneration. We have shown that not only noble metals but also non-noble Ni can be coupled with homogeneous catalysts for selective NADH regeneration. We have added the above discussions in the revised manuscript to avoid the detachment of the Section of chemoenzymatic reduction of acetophenone from the rest of the work.

Reviewer #2 (Remarks to the Author):

Yang and coworkers report a novel method for the hydrogenation/regeneration of NAD(P)⁺ to NAD(P)H with H₂ catalyzed by a tandem of supported metal nanoparticles (NPs, based on Ru, Rh, Pt, Pd and Ni) and an organometallic complex (M, based on Rh, Ir and Ru). The authors observe that the presence in the reaction media of both heterogeneous and homogeneous catalysts greatly improves the catalytic performance with selectivity up to > 99% and reaction rates about one order of magnitude higher than the corresponding activity for the metal NPs and M alone.

In addition, the tandem Ni/TiO₂ and M1 was also coupled to an enzyme (AKR) for the asymmetric hydrogenation of acetophenone reaching up to > 99% ee and at least 24 cycles of NADPH regeneration.

The work is interesting and competently executed. For these reasons, I recommend the publication of the paper in Nat. Commun. once the following minor revisions will be addressed.

Response: We sincerely appreciate your careful review and valuable suggestions. It really helps us a lot to improve our work. The questions are answered point-by-point as follows:

Question 1. I would suggest to the authors to determine the Turnover Frequency of the catalyst instead the reaction rate to better compare the activity of the studied complexes with the literature benchmarks.

Response: Thank you for your valuable suggestion. We calculated the specific activity in the kinetic region (normalized to all of the metals), and the results in this work together with the reported results were listed in Supplementary Table 1. To make it more readable, we added the specific activity in the revised manuscript in page 5.

Supplementary Table 1. Comparison of the results of NAD⁺ hydrogenation in this work and in literatures.

Catalyst	NAD ⁺ (mM)	T (°C)	H ₂ (atm)	pH	Sel. (%)	Specific activity (mol _{NAD⁺} mol ⁻¹ _{metal} h ⁻¹) ^a	Ref.
Ru/TiO ₂ @10TP-TTA & M1	1.5	37	20	8.7	>99	91.6	This work
	1.5	37	20	7.0	>99	90.7	
	1.5	37	1	8.7	>99	38.3	
	1.5	25	1	8.7	>99	13.6	
	15	37	20	9.4	92.0	47.2	
Pt/ana-450	1.5	37	1	8.7	63.4	19.9	1
Pt/SiO ₂	1.5	22	9	7.0	30	~512	2
Pt90Sn10/SiO ₂	0.1	22	9	7.0	~90	46.3	
[Ir(Cp [*])(4-(1H-pyrazol-1-yl-κN ²)benzoic acid-κC ³)(H ₂ O)] ₂ SO ₄	0.77	25	1	6.5	97 ^b	36	3
	0.77	25	1	7.0	97 ^b	~30	
	1.9	25	1	7.0	97 ^b	<10	

^aThe specific activity was calculated based on all of the metals used in reactions. ^bYield of NADH.

Question 2. The explanation of the reaction mechanism showed in Scheme 1 (page 11) is not clear and hard to read. I would suggest to the authors to improve that part of the manuscript.

Response: Thank you for your valuable suggestion. We have improved the discussion in the revised manuscript as follows: “With the aforementioned results, the following pathway may be involved in the hydrogenation of NAD⁺ with the homogenous-heterogeneous coupling system (Scheme 1). H₂ was first dissociated into H^{*} over Ru NPs. Subsequently, **M1** was reduced to [Cp^{*} Rh(I)(bpy)]⁰ using H^{*}, and the reduced Rh species was deposited on the surface of Ru NPs. Rh-H species was then formed after the coordination of proton to the metal center. Notably, the proton could be originated from both H₂O and the concomitant proton during the reduction of **M1**. Ultimately, the hydride was transferred to NAD(P)⁺ from [Cp^{*} Rh(bpy)H]⁺ though a ring-slipped mechanism⁴⁸, meanwhile **M1** was re-dissolved in the solution for the next catalysis cycle.”

Question 3. Do authors find any sign of speciation of NAD⁺ and NADH during their NMR isotopic studies (page 10 from line 10 on), as previously observed (see ACS Catal. 2017, 7, 7788–7796; this paper should be quoted)?

Response: Thanks for your kind reminding. In the ¹H NMR spectra of the reaction solution (Supplementary Fig. 6 and 7), the characteristic signal of NADH at 6.9 ppm could be observed. And a small peak at 7.1 ppm assigned to 1,6-NADH (by-product) was presented. No signal belonged to NAD⁺ could be observed because it was fully converted. We cited the article (ACS Catal. 2017, 7, 7788–7796) as reference 46 in the revised manuscript.

Question 4. Characterization data (such as dimensions and distribution of the NPs) of Ni@TiO₂ are not given. Please, provide it.

Response: Thank you for your suggestion. An improved HAADF-STEM image and the corresponding Ni size distribution of Ni/TiO₂ were added in the revised manuscript in Supplementary Fig. 18.

Supplementary Fig. 18. (A) HAADF-STEM image and (B) the corresponding Ni size distribution of Ni/TiO₂.

Question 5. The NMR spectra provided in the SI (in particular Figure S2 and S3) are unphased and of low quality. Please, improve the quality of the NMR spectra reported.

Response: Thank you for your suggestion. We have improved the quality of the NMR spectra. The revised spectra are presented in Supplementary Fig. 2, 5, 6 and 7.

Question 6. I would suggest to better elucidate the dependence of catalytic activity and selectivity on the thickness of TP-TTA and how it affects the formation of metallic Rh species.

Response: Thank you for your valuable suggestion. We further discussed the dependence of catalytic activity and selectivity on the thickness of TP-TTA in the revised manuscript in page 13 as follows: “In the coupling system, the competitive reduction of **M1** and NAD^+ on Ru NPs may determine the selectivity, which means that the high selectivity requires the fast hydrogenation rate of **M1** on Ru NPs but a slow one of NAD^+ . Although the net hydrogenation rate of **M1** on Ru NPs could not be determined directly, it could be roughly estimated from the NAD^+ conversion rate in the coupling system. As shown in Fig. 3d, both NAD^+ conversion rates on Ru NPs (r_1) and in the coupling system (r_2) were enhanced for Ru/TiO₂@TP-TTA with 1.8 nm TP-TTA overlayer, which was probably originated from the electron-rich surface of Ru as demonstrated by the in situ FT-IR results. However, it should be pointed out that r_2 was more pronouncedly enhanced than r_1 . For Ru/TiO₂@TP-TTA with 3.6 nm TP-TTA layer, r_2 decreased slightly, but a lowest r_1 was observed. This means that the selective NAD^+ hydrogenation route gradually dominates with the increase of the TP-TTA layer thickness, resulting in the enhanced selectivity.

The NAD^+ hydrogenation rate of Ru/TiO₂@TP-TTA & **M1** shows a volcano shape with the TP-TTA layer thickness (Fig. 3d), which is the combined effect of the hydrogenation activity for NAD^+ and **M1**. As compared with NAD^+ , **M1** has smaller molecular size. Considering the diffusion barrier, NAD^+ hydrogenation may be more severely suppressed on Ru/TiO₂@TP-TTA, providing more Ru sites for **M1** hydrogenation. This explains the higher activity of Ru/TiO₂@10TP-TTA & **M1** than Ru/TiO₂ & **M1**. The lower activity of Ru/TiO₂@10TP-TTA & **M1** than Ru/TiO₂@5TP-TTA & **M1** is caused by the lower surface exposure degree of Ru/TiO₂@10TP-TTA”.

We think that the TP-TTA layer has little effect on the formation of metallic Rh species and the reason that no metallic Rh was observed in the coupling system was discussed in page 15: “As mentioned above, no metallic Ru species were observed on Ru/TiO₂ by the treatment of Ru/TiO₂ & **M1** in H₂. This is possibly related with the fact that the reduced **M1** only adhered on the surface of Ru/TiO₂ and the lack of the free mobility prevent the aggregation of Rh species to form metallic Rh”.

Question 7. I would suggest obtaining more insights on why a negative correlation between the activity of metal NPs and the selectivity of the coupling system for Rh/TiO₂@5TP-TTA, Pd/TiO₂@5TP-TTA and Pt/TiO₂@5TP-TTA is observed.

Response: Thank you for your valuable suggestion. More insights on why a negative correlation between the activity of metal NPs and the selectivity of the coupling system for Rh/TiO₂@5TP-TTA, Pd/TiO₂@5TP-TTA and Pt/TiO₂@5TP-TTA have been discussed as “The selectivity of the coupling system shows a negative correlation with the activity of metal NPs for NAD⁺ hydrogenation. This is caused by the competitive reduction of NAD⁺ on **M1** and on metal NPs. The NADH selectivity in the coupling system is the sum up of the non-selective NAD⁺ hydrogenation on metal NPs and the selective NAD⁺ hydrogenation on reduced **M1**. To this end, it is reasonable to observe that higher NADH selectivity was obtained on metal NPs with less activity for NAD⁺ hydrogenation”.

Question 8. Enzymatic concentration is not given. Please, provide it.

Response: Thanks for your kind reminding. The enzymetic concentration (72.7 mg mL⁻¹) is given in the revised manuscript in page 21.

Question 9. Typos errors: page 6 a > 99%; SI fig. S6 H2 not N2.

Response: We thank the reviewer very much for the kind corrections. We have carefully checked throughout the manuscript and corrected the typo mistakes in the revised manuscript. The control experiment (Supplementary Fig. 5 in the revised SI) was carried out under D₂O/N₂ to preclude the H/D exchange between NADH and D₂O.

Question 10. The English of the manuscript is poor in many places making the meaning of some sentences little clear. For examples: 1) page 3, lines 12-13: “... it may help break though the plateau in catalysis”; 2) page 9, line 2: “... filtration reaction ...”; 3) page 10, line 9: “... the incorporation of proton into the ligand sphere ...” what does it mean?

Response: We thank the reviewer very much for the kind reminding. We have polished the English in the revised manuscript.

REVIEWER COMMENTS

Reviewer #1 (Remarks to the Author):

The revision from Wang et al. indeed supported my original assessment of this work. The questions risen were not well answered and some additional data even opens up more questions. I still think the quality of the revised manuscript is sub-standard since many of the issues being pointed out were not addressed.

Reviewer 1, Q1.

I have never hear of “continuous metal sites” in catalysis, maybe that’s a new thing. How did it affect NAD⁺ reduction? I don’t know. Even if that’s essential here, how can the authors guarantee that the TP-TTA not covers the TiO₂ surface (which apparently important) as well as these continuous metal sites? Also, even this response was poorly written. For example, “...the continuous metal sites which may cause unselective NAD⁺ adsorption should be avoided. To this end, M/TiO₂ catalysts were further coated with...” This seems to be translated by a software.

Q2.

While in Q1, the authors tried to justify the choice of the catalyst compositions, in the Table S1, it clearly showed that the results here were only incremental (if they are not worse than those in the literature). Take Ref 3 as an example, 22C, 1bar, pH 6.5, activity 36. This work, 25C, 1bar, pH7, pH 8.7 (which is less than ideal with many enzymes), activity 13.6.

I felt that the authors tried to ramp up the activity by using 20bar H₂, which is clearly impractical.

Q3.

The authors did make some improvement on the references but the revised version I received still have the same “Discussion” section, which was a conclusion, as the original submission. Nothing changed. There are still many references using XXX et al., while the other gave full lists. Just not good enough.

Q4.

The authors gave more data to justify their in depth investigation on Ru/TiO₂@10TP-TTA, with 3 additional entries in Table S3. This table is indeed all over the places with different time scales for entries. Why some were recorded in 5 mins while the other were in 6 or 8, I simply can’t understand. For example, entry 7 was chosen over entry 10 but it was carried out for 8 mins. What about the result in 6 mins? Similar to 7?

Q5.

The response indeed supported my original assessment here. The Ni/TiO₂ work was not linked to the Ru/TiO₂@10TP-TTA work. As the authors pointed out, “... In this work, Ru/TiO₂@10TP-TTA was firstly chosen as a model catalyst to couple, with enzyme because of its high NADH selectivity. However, it was found that Ru/TiO₂@TP-TTA could directly catalyze the hydrogenation of acetophenone, which results in the decrease in enantioselectivity.” After spending 16 pages to read about the development of

Ru/TiO₂@TP-TTA and now the authors told us that was actually not very good! And they also abandon M1 in favour of M2. The catalytic system was totally different.

Reviewer #2 (Remarks to the Author):

The answers of authors and the modifications that they introduced in the manuscript addressed all points I addressed. For those reasons I recommend its publication in Nature Communications as it is.

Responses to Reviewer 1

The revision from Wang et al. indeed supported my original assessment of this work. The questions risen were not well answered and some additional data even opens up more questions. I still think the quality of the revised manuscript is sub-standard since many of the issues being pointed out were not addressed.

Response: We sincerely appreciate your careful review and valuable suggestions. We have carefully revised our manuscript according to your comments. Our work mainly focuses on the synergy of metal nanoparticles and organometallic complex in NAD(P)H regeneration via relay hydrogenation. As we mentioned in the Introduction Section, homogenous and heterogeneous catalysts were applied separately and rarely catalyzed a target reaction synergistically. In this work, we make these two types of catalysts communicate with each other in a controllable manner for NADH regeneration. The format of this manuscript was prepared carefully according to the GUIDE TO FORMATTING ARTICLES of Nature Communications (details see response to Q3). We think that the novelty and quality of this work warrant publication in Nature Communications. The questions are answered point-by-point as follows:

Q1.

I have never hear of “continuous metal sites” in catalysis, maybe that’s a new thing. How did it affect NAD⁺ reduction? I don’t know. Even if that’s essential here, how can the authors guarantee that the TP-TTA not covers the TiO₂ surface (which apparently important) as well as these continuous metal sites?

Also, even this response was poorly written. For example, “...the continuous metal sites which may cause unselective NAD⁺ adsorption should be avoided. To this end, M/TiO₂ catalysts were further coated with...” This seems to be translated by a software.

Response: Thanks for your kind reminding. For the supported metal nanoparticles, there are different types of metal sites as shown in Figure 1. The “continuous metal sites” was often used to describe the atoms in facet (Chem. Rev. 2020, 120, 683–733). The nicotinamide ring of NAD⁺ tends to adsorp on these metal sites by a parallel configuration, which is not beneficial for the formation of 1,4-NADH. The coverage of these continuous metal sites could increase the 1,4-NADH selectivity.

We agree with the reviewer that the TP-TTA covers both the the TiO₂ surface as well as these continuous metal sites. Because TiO₂ cannot activate the H₂ under current conditions, the surface

coverage of TiO₂ with TP-TTA layer has little influence on the catalytic performance of supported metal NPs.

Figure 1. Illustration of the geometric structure of supported metal nanoparticles.

In order to avoid the misleading, we changed the sentence “...the continuous metal sites which may cause unselective NAD⁺ adsorption should be avoided. To this end, M/TiO₂ catalysts were further coated with...” to “The formation of side products in metal NPs catalyzed NAD⁺ hydrogenation may be caused by the NAD⁺ adsorption on facet metal atoms by a parallel configuration. Our previous report show that TP-TTA polymer could efficiently cap the connected metal atoms in facet⁴². Herein, ...” in the revised manuscript.

Q2.

While in Q1, the authors tried to justify the choice of the catalyst compositions, in the Table S1, it clearly showed that the results here were only incremental (if they are not worse than those in the literature). Take Ref 3 as an example, 22C, 1bar, pH 6.5, activity 36. This work, 25C, 1bar, pH7, pH 8.7 (which is less than ideal with many enzymes), activity 13.6.

I felt that the authors tried to ramp up the activity by using 20bar H₂, which is clearly impractical.

Response: We did not agree with the Reviewer that our results were only incremental as compared with the results reported in the literature. For NAD⁺ regeneration, the selectivity is the most important parameter to evaluate the catalytic performance of a catalyst. As listed in Table S1, our catalyst gave >99% NADH selectivity, higher than the reported catalysts. Though [Ir(Cp^{*})(4-(1H-pyrazol-1-yl-κN²)benzoic acid-κC³)(H₂O)]₂SO₄ can catalyse the NAD⁺ hydrogenation in milder conditions, the NADH selectivity is lower than our catalyst. In addition, it's known that the selectivity of NAD⁺ hydrogenation is sensitive to the reaction conditions. The higher

temperature and pressure may lead to a lower selectivity. However, our catalyst presents excellent selectivity at different conditions, which is one of the merits for practical applications.

Most oxidoreductases are stable with pH in the range of 7-8 (Biochimica et Biophysica Acta, 1988, 950(1): 54-60.). Our catalyst can work well at this pH range. Furthermore, the NAD⁺ hydrogenation could be performed after biocatalysis. In industry, the hydrogenation reaction was often performed at H₂ pressure higher than 1 bar. And the hydrogenation reaction with 20 bar H₂ can be easily applied in industry. Thus, the conditions used in our work will not cause difficulties in practical applications.

In order to avoid the misunderstanding, the sentence “ ... superior to most reported homogeneous and heterogeneous catalysts (Supplementary Table 1)” was changed to Ru/TiO₂@10TP-TTA and **M1** is more selective than the reported homogeneous and heterogeneous catalysts... in the revised manuscript.

Q3.

The authors did make some improvement on the references but the revised version I received still have the same “Discussion” section, which was a conclusion, as the original submission. Nothing changed. There are still many references using XXX et al., while the other gave full lists. Just not good enough.

Response: We sincerely appreciate your careful review. According to the GUIDE TO FORMATTING ARTICLES of Nature Communications, a section heading of Conclusion is not permitted, while Discussion is generally adopted although it is optional. To avoid the ambiguity, we delete it in the revised manuscript.

Nature Communications uses standard Nature referencing style. All authors should be included in reference lists unless there are six or more, in which case only the first author should be given, followed by 'et al.'.

Q4.

The authors gave more data to justify their in depth investigation on Ru/TiO₂@10TP-TTA, with 3 additional entries in Table S3. This table is indeed all over the places with different time scales for entries. Why some were recorded in 5 mins while the other were in 6 or 8, I simply can't understand. For example, entry 7 was chosen over entry 10 but it was carried out for 8 mins. What about the result in 6 mins? Similar to 7?

Response: We thank the reviewer for the comments. We measured the NAD⁺ conversion with the reaction time. The reaction time to reach the (nearly) full conversion of NAD⁺ for the coupling system is listed in Table S3. Considering that the NAD⁺ conversion may influence the NADH selectivity, the reaction time for the (nearly) full conversion of NAD⁺ for the coupling system was carefully chosen to make the reasonable selectivity comparison. When the reaction was carried out for 6 mins for entry 7, the conversion is 88.1%. Therefore, the result of 8 mins was used in the table. We added the above discussions in the revised manuscript in page 15.

To make it more clarify, the results of supported metal NPs are listed together in Supplementary Table 3 and the results for the coupling system are listed in the order of the TP-TTA layer thickness in the revised manuscript.

Q5.

The response indeed supported my original assessment here. The Ni/TiO₂ work was not linked to the Ru/TiO₂@10TP-TTA work. As the authors pointed out, "... In this work, Ru/TiO₂@10TP-TTA was firstly chosen as a model catalyst to couple, with enzyme because of its high NADH selectivity. However, it was found that Ru/TiO₂@TP-TTA could directly catalyze the hydrogenation of acetophenone, which results in the decrease in enantioselectivity." After spending 16 pages to read about the development of Ru/TiO₂@TP-TTA and now the authors told us that was actually not very good! And they also abandon M1 in favour of M2. The catalytic system was totally different.

Response: We do not agree with the reviewer that the catalytic system was totally different. As pointed by the reviewer, we spent 16 pages to describe the synergy of supported metal NPs and homogeneous catalysts in the H₂-driven NADH regeneration. We also demonstrated that the synergy effect was a general phenomenon for Ru, Rh, Pd, Pt and Ni NPs together with different types of metal complexes. This is the main novelty and content of our work. The in situ NADH regeneration in biocatalysis is not the main focus of this work and in fact the catalytic system of Ni/TiO₂ and **M2** for in situ NADH regeneration is in the network of the catalytic system reported in our work.

We started our story with Ru/TiO₂@TP-TTA and **M1** is due to the following reasons: (1) Ru/TiO₂@10TP-TTA & **M1** gave the highest NADH selectivity. (2) **M1** is the most often used metal complex for NADH regeneration via catalytic hydrogenation, photocatalysis and electrocatalysis and the catalytic mechanism of **M1** is well investigated. This will help to elucidate the hydride relay from supported metal NPs to metal complexes.

On the consideration of the reviewer's comments, the 'chemoenzymatic reduction' section was moved to **Supplementary information** in the revised manuscript.

Responses to Reviewer 2

The answers of authors and the modifications that they introduced in the manuscript addressed all points I addressed. For those reasons I recommend its publication in Nature Communications as it is.

Response: Thank you for your valuable suggestions and recommendation.

REVIEWERS' COMMENTS

Reviewer #1 (Remarks to the Author):

Since the authors decided not to address a number of issues that I pointed out, I would leave it to the editor to decide the outcome of this revision.

Responses to Reviewer 1

Since the authors decided not to address a number of issues that I pointed out, I would leave it to the editor to decide the outcome of this revision..

Response: Thank you for your review and comments.